# Social bonds decrease epigenetic age in male bottlenose dolphins
Livia Gerber [1,2] ✉, Katharina J. Peters [3,4,5], Stephanie L. King [6,7], Simon J. Allen [5,6,7],
Richard C. Connor [8,9], Owen Forbes [10], Kathryn G. Holmes [7,11], Anna M. Kearns [2], Erik P. Willems [5],
Michael Krützen [5,7] & Lee A. Rollins [1]

Ageing is a universal process characterised by the deterioration of functional traits over an individual's lifespan. Differing degrees of age-related decline between individuals of the same chronological age suggest varying rates of ageing. Identifying factors influencing these inter-individual differences in 'biological age' is central to understanding ageing. In social mammals, social variables affect lifespan and are therefore likely to affect biological age. In Shark Bay, Western Australia, male bottlenose dolphins forge persisting intrasexual social bonds that affect their reproductive success and, therefore, their evolutionary fitness. We investigate the relationship between cumulative social bond strength, group size, and biological age of individual male dolphins in this population. Biological age is inferred using a species-specific epigenetic clock, the current gold standard for such an inference. We find a significant negative effect of cumulative social bond strength and a significant positive effect of group size. This implies that the negative effect of social bonds on epigenetic age cannot be attributed solely to group-living, but to benefits of the social bonds themselves. As established in humans, we find that the strength of social relationships affects epigenetic age, indicating that sociality may be linked to biological ageing more broadly across social mammals.

Ageing is an inevitable process in most forms of life, characterised by the gradual deterioration of functional attributes[1]. Yet, not all individuals of the same species age at the same rate, as evidenced by considerable variation in the physical decline of individuals of the same chronological age, i.e. the time that has elapsed since birth[2,3]. Such inter-individual differences in ageing can have major consequences for evolutionary fitness in species with slow life histories, where lifespan is one of the major sources of differential fitness[4]. The identification of variables contributing to the rate of ageing is central to evolutionary biology because life history theory revolves around finding optimal solutions in the allocation of resources to somatic (growth and maintenance) versus reproductive effort.

Social factors are known to have a positive influence on lifespan, particularly in social mammals (reviewed in ref. 5). Social bonds, defined here as persisting affiliative relationships between individuals, increase survival in

rhesus macaques (*Macaca mulatta*)[6] and blue monkeys (*Cercopithecus mitis stuhlmanni*)[7]. In killer whales (*Orcinus orca*), big horn sheep (*Ovis canadensis*), and rock hyraxes (*Procavia capensis*), an individual's connectivity to others within their social network also improves their survival[8–10]. The positive effect of having strong social bonds on survival and lifespan can occur independently of environmental conditions, as shown in female baboons (*Papio cynocephalus*)[11], and can exceed that of harmful behaviours such as smoking and alcoholism in humans[12].

To date, most studies have investigated the influence of social factors on lifespan (chronological age). However, for a comprehensive understanding of the interplay between social factors and age, we also need to consider biological age. Biological age reflects an individual's physiological age and serves as an indicator of overall health and ageing status[2]. Consequently, biological age can differ from chronological age within an

[1]Evolution & Ecology Research Centre, School of Biological, Earth and Environmental Sciences, University of New South Wales, Sydney, NSW, Australia.
[2]Australian National Wildlife Collection, National Research Collections Australia, CSIRO, Canberra, ACT, Australia. [3]Marine Vertebrate Ecology Lab, Environmental Futures, School of Science, University of Wollongong, Wollongong, NSW, Australia. [4]Securing Antarctica's Environmental Future, University of Wollongong, Wollongong, NSW, Australia. [5]Evolutionary Genetics Group, Institute of Evolutionary Anthropology, University of Zurich, Zurich, Switzerland. [6]School of Biological Sciences, University of Bristol, Bristol, UK. [7]School of Biological Sciences, Oceans Institute, University of Western Australia, Crawley, WA, Australia. [8]Biology Department, University of Massachusetts Dartmouth, North Dartmouth, MA, USA. [9]Institute of Environment and Department of Biological Sciences, Florida International University, North Miami, FL, USA. [10]National Research Collections Australia, CSIRO, Canberra, ACT, Australia. [11]Present address: Brookfield Zoo Chicago's Sarasota Dolphin Research Program, c/o Mote Marine Laboratory, Sarasota, FL, USA. ✉e-mail: livia.gerber@csiro.au

individual and between individuals of the same chronological age. Advances in molecular biology over the last decade permit the estimation of biological age via epigenetic clocks[13–15]. Epigenetic clocks are based on the identification of DNA sites that gain or lose methyl groups in a predictable manner over the course of a species' lifespan[13]. Although initially calibrated in humans, the development of a universal mammalian clock has demonstrated that these clocks can be generated from all trialled tissues across mammals[14], and potentially even across vertebrates[16]. By providing a reliable biomarker for biological age, epigenetic clocks have opened up new avenues for comparative ageing research across diverse taxa, including the effect of environmental, genetic, and phenotypic variables on biological ageing.

In humans, age estimates derived from DNA methylation data ('epigenetic age') have been widely used to infer how environmental and genotypic variables affect biological age[13,17]. Various factors such as pollution, depression, and male sex have been linked to an increase in epigenetic age (reviewed in ref. 18). Conversely, perceived social support and contact frequency with others are associated with a decrease in epigenetic age in humans[19–22]. This suggests that social factors may not only affect lifespan (i.e., maximum chronological age), but also the biological ageing process. Only a few studies have investigated the effect of social variables on the rate of ageing outside of humans. The results have been mixed: while high social status was linked to accelerated ageing in male adult baboons (*Papio cynocephalus*), social bond strength in female baboons did not explain variation in ageing[23]. Similarly, pair-bonding in prairie voles (*Microtus ochrogaster*) did not appear to affect the rate of ageing[24]. Given these mixed results across species and the limited number of studies, the relationship between social bonds and biological ageing requires further investigation. This is crucial not only for comparative purposes, but also to shed light on how social factors may buffer against stress, and thus protect against accelerated ageing, as well as to gain insights into the evolutionary trade-offs between social behaviour, resource allocation, and longevity.

The population of Indo-Pacific bottlenose dolphins (*Tursiops aduncus*, referred to as 'dolphins' hereafter) in Shark Bay provides an excellent natural study system to address questions on the interplay between sociality and biological ageing. Over 40 years of individual-level behavioural data allow reliable social network inference, while epigenetic clocks can be calibrated using genetic samples and demographic information. The approximately 1500 dolphins in our study population have a lifespan of over 40 years and live in an open social network with a fission-fusion grouping pattern, overlapping home ranges, year-round residency and bisexual philopatry[25–28]. Moreover, males in this population forge persisting social bonds in a multi-level alliance system[29,30] that is considered one of the most complex social systems outside of humans[31]. The formation of these male alliances begins in the late juvenile period and crystallises more than a decade later when individuals reach social maturity[32]. Previous research found a positive correlation between male social bonds and reproductive success, where individuals with strong social bonds with alliance partners sire more offspring than less socially connected males[29].

In this study, we investigated whether male dolphins that were more connected to other males tended to be biologically younger, i.e., have a younger epigenetic age than expected for their chronological age, when compared to less social males. To achieve this, we employed a species-specific epigenetic clock while controlling for chronological age to isolate the effect of social variables on biological age[33]. Controlling for chronological age is crucial because social factors can change over an animal's lifespan, and thus with chronological age[34,35]. However, in Shark Bay dolphins, an individual's inherent social traits, e.g., highly versus less social, are usually stable across their lifetime[36]. We also investigated the effect of mean male group size and variation in intrasexual social bond strength on epigenetic age. This allows us to test if mean male group size alone or social bonds themselves affected epigenetic age, as well as whether males that invest more equally in social bonds with other males not only sire more offspring[29] but are also epigenetically younger. By applying an epigenetic clock in a natural system outside of the primate lineage to investigate how sociality is linked to molecular ageing, this study expands our knowledge of the evolution of

sociality and longevity. Moreover, by furthering our understanding of this complex relationship, we may gain invaluable insight into the fundamental biological processes that link social behaviour, epigenetics, and lifespan, with far-reaching implications for both evolutionary biology and health.

## Results
### Epigenetic clock calibration

To calibrate our species-specific epigenetic clock, we identified 68 individual dolphins of both sexes whose chronological ages were known with an accuracy of ±1 year and for whom sufficient skin tissue was available for epigenetic analyses. A subset of 18 individuals was sampled at multiple ages, bringing the total available data for epigenetic clock calibration to 90 samples. We used elastic net regression models with alpha set to 0.5 for clock generation, entering the 29,813 probes of the HorvathMammalMethylChip40[37] mapping to the *T. aduncus* genome as explanatory variables, and chronological age as the response variable. The epigenetic clock we used here, constructed with the leave-one-individual-out cross validation (LOIOCV) approach (median absolute error MAE = 2.0 years and $r$ = 0.86, Fig. 1 and Supplementary Fig. 1) performed similarly to our epigenetic clock using a leave-one-out cross validation (LOOCV) approach (MAE = 2.1 years and $r$ = 0.86, details are provided in ref. 38). Sex, chronological age when sampled, sampling year, and, where available, social metrics on the individuals included in our calibration dataset are provided in Supplementary Data 1. Genetic information (methylation proportion) can be found in Supplementary Data 2.

### Statistical analyses of relationships between social bonds and epigenetic age

To investigate if epigenetic age is affected by social connectivity, we identified 38 focal males in the dataset used for epigenetic clock calibration. The focal males were between 0 and 29 years old when sampled, and all were seen at least 15 times until their first (or only) skin tissue sample was collected, permitting reliable inference of their social networks[39]. A subset of ten individuals was sampled at multiple ages, resulting in a total of 50 samples available for statistical analyses (see Supplementary Data 1 for information on the focal individuals and samples).

Based on the lowest Akaike Information Criterion (AIC) value, we chose the most supported linear mixed-effects model (LMM), which included the following predictors: cumulative social bond strength, male group size, and chronological age as fixed effects and year sampled and dolphin ID as random effects. Full details on model comparison are included in the Supplementary Information (Supplementary Tables 1 and 2). The selected LMM revealed a statistically significant negative effect of cumulative social bond strength, i.e., social connectivity, on epigenetic age (Table 1 and Fig. 2, $P$ = 0.012, $b$ = -9.75), and a significant positive relationship with mean male group size in male dolphins ($P$ = 0.013, $b$ = 0.90). A 1 standard deviation (0.17) increase in cumulative social bond strength is associated with a 1.69 years lower epigenetic age (95% CI: −3.03, −0.35), while a 1 standard deviation (1.44) increase in male group size is associated with 1.33 years higher epigenetic age (0.27, 2.40), while accounting for the other fixed and random effects included in our main model. Group size and cumulative bond strength were neither collinear nor strongly correlated with each other (see Supplementary Table 3).

Given the possibility of more complex causal pathways than those captured in our principal analyses, and the 'table 2 fallacy' risk of potentially over-interpreting regression coefficients from mixed-effects models[40], we conducted a mediation analysis to assess whether social bond strength mediates the relationship between male group size and epigenetic age acceleration. Our mediation analysis (see Supplementary Information and Supplementary Table 4) found no significant mediation effects. This suggests that while both group size and social bonds independently influence epigenetic ageing, the causal pathway is not well captured by a simple mediation relationship. Future studies with larger sample sizes could employ more sophisticated analytical frameworks, such as structural equation modelling[41] to disentangle the potentially complex causal

interrelationships between social structure, bond formation, and biological ageing processes.

It is important to note that while the raw data shows a positive relationship between social bond strength and epigenetic age (Supplementary Fig. 2), our model, which controls for chronological age, reveals a negative relationship between predicted epigenetic age and social bond strength, as evident in the partial residuals plot (Fig. 2). This apparent contradiction is due to chronological age acting as a confounding variable. By controlling for chronological age, we remove its confounding effect, uncovering the underlying negative relationship between social bond strength and epigenetic age. Thus, our results suggest that among individuals of the same chronological age, those with stronger social bonds tend to have a lower epigenetic age. This finding was also reproduced when restricting the dataset to males whose chronological ages were known with an accuracy of ±6 months ($P = 0.037$, $b = -11.25$), to adult males (≥14 years) with established alliances ($P = 0.03$, $b = -12.66$), and when individual age acceleration values, the residuals of a linear regression of epigenetic age on chronological age (AgeAccel, Supplementary Fig. 3) were specified as the response variable ($P = -0.015$, $b = -9.84$, detailed model results for these analyses are provided in Supplementary Tables 5–7 and Supplementary Fig. 4). AgeAccel represents how much older or younger an individual is epigenetically compared to what would be expected based on their chronological age[15].

Interestingly, mean male group size had a significant positive effect on epigenetic age in the LMM as well as when AgeAccel was the response variable ($P = 0.015$, $b = 0.92$, Table 1, Fig. 2, Supplementary Table 7, and

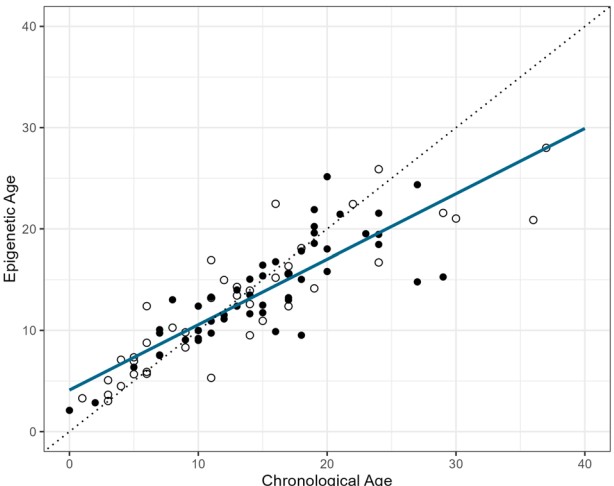

**Fig. 1 | Epigenetic age versus chronological age for Shark Bay Indo-Pacific bottlenose dolphins.** Epigenetic ages were calculated for 68 individuals, totalling 90 samples, using elastic net regression models with leave-one-individual-out cross validation (LOIOCV). The regression line is shown in blue. The dotted diagonal line indicates epigenetic age = chronological age. Filled circles denote the subset of 50 samples across the entire lifespan that are included in our main analysis on the interplay between social bonds and epigenetic age. Empty circles indicate samples that were only included in the dataset for clock generation.

Supplementary Fig. 4). This suggests that males observed in larger groups are characterised by higher epigenetic ages, and thus age faster. However, this finding was not replicated when our input data was restricted to males for whom age was known with higher accuracy ($P = 0.14$, $b = 0.75$) or adult males only ($P = 0.19$, $b = 0.78$, detailed results in Supplementary Tables 5 and 6). This might be caused by a lack of statistical power due to smaller sample size.

The coefficient of variation (CV) in social bond strength was not included in our final model, and thus seems to be of little explanatory power for epigenetic age, despite its significant relationship with reproductive success[29]. The findings obtained from applying seven previously described epigenetic clocks did neither contradict nor confirm our main findings in that only chronological age or no variable significantly correlated with epigenetic age estimates. We describe and discuss the findings and the additional epigenetic clocks applied to our data in more detail in the Supplementary Information (Supplementary Note and Supplementary Figs. 6–12 and Supplementary Tables 8–14).

## Discussion
Here, we show that social bonds are linked to decreased epigenetic ages in male Indo-Pacific bottlenose dolphins and thus may decrease the rate of ageing. Individual male dolphins with higher cumulative same-sex social bond strengths, a metric reflecting social connectivity, were characterised by lower epigenetic age estimates compared to socially less connected individuals. Given that epigenetic age reflects biological age[13], individuals with more and/or stronger social bonds appear to age more slowly biologically than they do chronologically. A similar pattern has been found in humans: social contact frequency and perceived social support correlated negatively with epigenetic age in adults over 50[20,21]. Similarly, paramedics with more social support had lower epigenetic ages before and after trauma exposure than their colleagues with less social support[19]. We observe a similar pattern in an evolutionarily distant mammalian lineage, so it appears likely that social bonds affect not only lifespan[5] but also epigenetic age, and the biological ageing process more broadly in highly social mammals. This convergence across two distant taxa underscores the potential universality of the link between sociality and ageing. However, while these findings are compelling, more studies are needed to fully understand the complex relationship between ageing and sociality.

Social factors and health are highly interlinked in social mammals; social adversity increases the risk of multiple diseases and death (reviewed in ref. 5). Moreover, laboratory studies revealed that social isolation alters regulation of the immune system, increasing inflammation[42], as well as neuroendocrine signalling[43,44]. Inflammation and chronic stress both accelerate ageing[45–47]. Hence, the relative lack of social bonds may accelerate ageing by increasing stress levels in social mammals. The male dolphins of Shark Bay are highly social, with allies relying on their social partners for successful reproduction[29]. Therefore, social isolation is a likely source of stress and may, in turn, accelerate ageing. Indeed, social instability has been shown to impair dolphin health[48], and the loss of closeness centrality (a measure indicative of social network position) was recently found to precede death in male dolphins[49].

## Table 1 | Model summary

| Explanatory variable | Parameter estimates | Standardised parameter estimates (95% CI) | p value |
|---|---|---|---|
| Epigenetic age ($N_{Samples} = 50$, $N_{Ind} = 38$) | | | |
| Intercept | 1.75 (−1.73, 5.23) | | 0.32 |
| Cumulative social bond strength | −9.75 (−17.25, −2.26) | −1.69 (−3.03, −0.35) | **0.012** |
| Mean male group size | 0.90 (0.20, 1.61) | 1.33 (0.27, 2.40) | **0.013** |
| Chronological age | 0.74 (0.55, 0.93) | 4.76 (3.50, 6.02) | **<0.0001** |

Summary statistics (regression coefficients and p values) obtained from an LMM that expresses epigenetic age as a function of social connectivity (cumulative social bond strength) and mean male group size, controlling for a male's chronological age and including sampling year and individual ID as random effects. p values in bold are considered significant (<0.05).

**Partial Residual Plots for Fixed Effects**

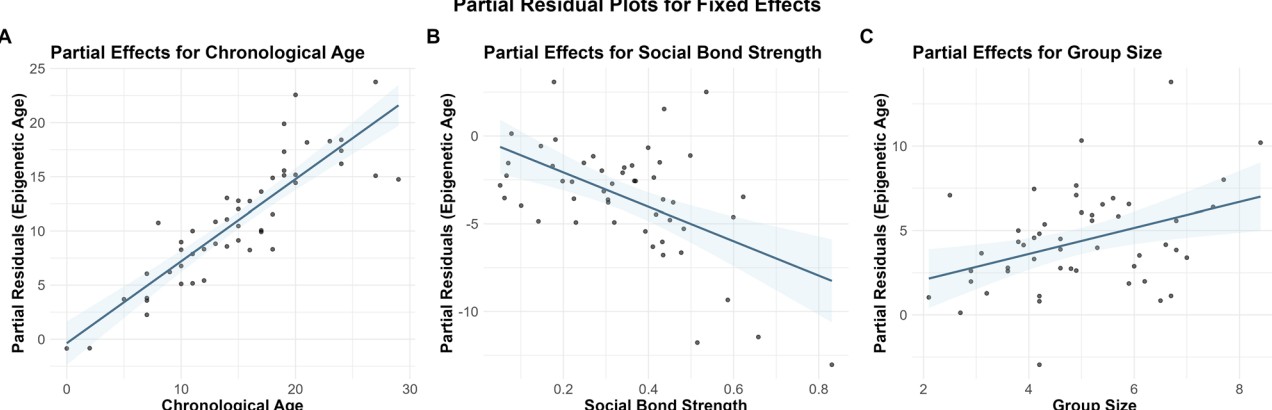

**Fig. 2 | Partial effects of chronological age, social bond strength, and group size on epigenetic age in bottlenose dolphins.** Each panel shows partial residuals (black dots) and model predictions (blue lines with 95% confidence intervals) representing the relationship between one predictor and epigenetic age while controlling for the effects of all other variables in the model. **A** Positive partial effect of chronological age on epigenetic age. **B** Negative partial effect of social bond strength on epigenetic age. **C** Positive partial effect of group size on epigenetic age. Partial residuals were calculated by removing the fitted effects of all other predictors from the response variable, ensuring that both the plotted points and regression lines represent the same statistical relationships. Random effects for individual dolphin ID and year sampled were included in the model but are not visualised here. See Supplementary Fig. 5 for an alternative visualisation showing raw observed data values plotted with model-predicted relationships.

Mean male group size showed a positive correlation with epigenetic age in the Shark Bay dolphin population. This suggests that males in larger groups are characterised by higher biological ages compared to those in smaller groups. Thus, the negative effect of social bonds on epigenetic age observed here cannot be attributed solely to group-living but to benefits of the social bonds themselves. The positive effect of mean male group size on epigenetic age could also arise from the costs of intragroup resource competition. Female dolphins only produce a single offspring approximately every four years[50], resulting in a highly skewed operational sex ratio, forcing males to compete over females. The number of male associates likely relates to the number of allies a male has, which, in Shark Bay, varies over an order of magnitude[31]. While having more allies is beneficial in competition with rival groups, more social relationships include more relationship uncertainty and, likely, stress. Our results suggest that, for a given number of allies, those with stronger social bonds will lead less stressful lives than those with weaker bonds. These interactions may accelerate biological ageing, as recently has been demonstrated in baboons via an epigenetic clock[23]. Furthermore, despite offering protection against predators, e.g., tiger sharks (*Galeocerdo cuvier*) that are highly prevalent in Shark Bay[51], group-living can increase disease risk[52]. In humans, various diseases have been linked to accelerated ageing, such as Covid-19[53], HIV-1[54] and cancer[55]. Additional studies are required to disentangle the sheltering effect of living in larger groups that may decelerate ageing from disease risk that is linked to accelerated ageing.

Interactions between the social environment, DNA methylation, and gene regulation have been described in various social mammal species across multiple tissues[56]. In hyaenas, for example, social connections and received maternal care in early life influenced global blood DNA methylation in adulthood[57], while methylation patterns of gut epithelial cells were associated with social status[58]. Moreover, social stress was correlated with DNA methylation in rat (*Rattus norvegicus*) hippocampal samples[59] and placental samples from rhesus macaques (*Macaca mulatta*)[60]. We hypothesise that a male dolphin's social environment similarly influences DNA methylation, and thus gene regulation of physiological processes that affect overall health. These processes are likely to impact life history.

As previously established in humans, we find that social bonds slow the rate of ageing in male bottlenose dolphins. Our study indicates that social bonds not only affect survival and lifespan but also the ageing process itself. While our findings do not permit definitive conclusions on the causality between ageing and social bonds, we can hypothesise about several potential mechanisms for this relationship. Social bonds could decelerate the ageing process through various pathways. For instance, strong social bonds might reduce stress as found in various social animals[61–63]. Social bonds can reduce stress by improving access to resources via cooperation, or reducing energy expenditure in foraging or defence activities[5]. This stress reduction could lead to decreased inflammation and oxidative stress, both of which are associated with accelerated ageing. Alternatively, individuals that age more slowly might be better equipped to invest and maintain social bonds. Slower ageing is likely to be associated with better physical condition, allowing individuals to participate in social interactions, including the various cooperative behaviours displayed by male dolphin alliances. These mechanisms are not mutually exclusive and could operate in a feedback loop, where strong social bonds and slower ageing reinforce each other over time, improving an individual's overall health. Future research could shed light on the mechanisms underlying the relationship between ageing and sociality, and this might be best achieved in an experimental system.

By applying an epigenetic clock in a free-ranging animal population, we not only demonstrate the importance of long-term study populations in studying the evolution of sociality and ageing but also shed light on the relationship between the two in highly social taxa. Our findings open new avenues for future research into the connections between sociality, ageing, and health.

## Methods

### Behavioural data and tissue sample collection

This study is based on long-term behavioural and genetic data collected on the population of Indo-Pacific bottlenose dolphins (*Tursiops aduncus*) off Monkey Mia in the eastern gulf of Shark Bay, Western Australia[30,64]. Standardised behavioural observations in the form of 5-min surveys of opportunistically encountered groups have been conducted on this population since 1984[30], with tissue samples for genetic analyses being collected from 1994 onwards[65]. Upon collection, tissue samples were stored in RNAlater (Thermo Fisher Scientific) or saturated NaCl and 20% dimethylsulfoxide (DMSO) at −20 °C in the field and −80 °C in the lab. Group composition during observational surveys was assessed using standard photo-identification methods based on inter-individual dorsal fin differences[66].

We have complied with all relevant ethical regulations for animal use, and our study was done in compliance with the animal ethics policies of the University of Zurich, University of Bristol, and University of Western Australia. Permits for the scientific use of animals were obtained from the Department of Biodiversity, Conservation and Attractions, Western Australia. LG, SLK, SJA, RCC, KGH, and MK were involved in fieldwork.

## Chronological age

For the purposes of this study, we focused on individuals whose chronological ages were known with an accuracy of ±1 year or less. This is because accurate chronological age estimates are required for epigenetic clock calibration. We assigned an individual's birth date based on their first sighting as a calf. Calf age is estimated using behavioural and morphological criteria such as surfacing patterns, presence of foetal folds, and the last sighting of the mother before the birth[67].

## Epigenetic age

We extracted DNA using a Quick-DNA Miniprep Plus Kit (Zymo) and subsequently purified the DNA with a DNA Clean & Concentrator Kit (Zymo). We produced DNA methylation data using the HorvathMammalMethylChip40, a customised Infinium array for mammal species[37] targeting roughly 37,000 CpG sites. The same data was used for a previous study[38]. For epigenetic clock calibration, we only included probes mapping to the *T. aduncus* genome assembly (GenBank assembly accession: GCA_003227395.1[68]). We identified the probes on the HorvathMammalMethylChip40 that map to our species' genome using Bowtie2 with a quality score of 10[69]. To estimate an individual dolphin's epigenetic age, we recalibrated our species-specific epigenetic clock that was calibrated on the Shark Bay dolphins[38] with one minor modification: instead of using elastic net regression models with a leave-one-out cross validation (LOOCV) to generate epigenetic age estimates, we relied on leave-one-individual-out cross validation (LOIOCV) here. This approach was deemed more appropriate for the current study because we had multiple samples available for a subset of individuals, and thus some of the data were non-independent. We performed elastic net regression and LOIOCV using the 'tidymodels' R package v1.2.0[70], setting the engine to 'glmnet'.

We also applied seven previously described epigenetic clocks to our samples to infer the robustness of our findings across a variety of epigenetic clocks[14,71–73]. To obtain epigenetic age estimates for these additional clocks, we used the 'MammalMethylClock' R package v1.0.0[74]. Other R packages used in our analyses included 'ggpubr' v0.6.0[75], 'ggplot2' v3.5.1[76], 'glmmTMB' v1.1.10 [77], gridExtra' v2.3[78], 'car' v3.1-3[79], 'lme4' v1.1-35.5[80], 'lmerTest' v3.1-3[81], and 'mediation' v4.5.0[82].

## Sex

We confirmed the male sex of all focal individuals included in our study using a multiplex approach of sex chromosome-specific loci ZFX and SRY, as described in ref. 38.

## Social connectivity

Social bond strength in dolphins can be measured based on patterns of association[32]. Here, we estimated dyadic social bond strength by calculating the Simple Ratio Index (SRI) using the 'asnipe' R package v1.1.17[83]. The SRI takes values between zero and one, indicating the proportion of surveys in which two animals are seen together. Hence, individuals that are never observed together have an SRI of 0, while individuals that are never seen apart are characterised by an SRI of 1. We calculated the SRI values of focal males by including surveys from the first time a male was observed up to and including the year it was sampled (i.e. the point in time at which epigenetic age was determined). For repeatedly sampled individuals, this meant that social connectivity was calculated multiple times, once per sampling point. Only association data recorded in the first 5 min of a survey were used to ensure association measures were comparable across surveys. Resightings, where the same group was encountered repeatedly within 2 h, were excluded. We also removed all surveys in which the predominant behavioural activity was foraging (defined based on inter-individual spacing, relative orientation, dive type, and direct observations of prey or feeding), because animals tend to loosely aggregate in large groups at the same foraging patch but are not necessarily associating[31]. We only included males that had been seen at least 15 times across the study period, from an individual's first sighting up until it was sampled[39]. For all animals with a minimum of 15 sightings, the timespan, and thus age range over which the social variables

are calculated are provided in Data S1. We subsequently used the SRI values to calculate cumulative social bond strength (also referred to as 'weighted degree' or 'normalised node strength'), a metric employed in social network analysis to reflect a male's overall connectivity to other males. An individual's node strength is defined as the sum of its SRI values, i.e. its cumulative social bond strength. However, because not all individuals were sampled in the same year and because a different number of individuals were seen each year, we normalised node strength by dividing each male's node strength by the maximum node strength calculated across all males seen 15 times or more up until the sample of the focal male was taken. Therefore, the cumulative social bond strengths of each data point reflect the timespan between any focal male's first sighting until the year(s) it was sampled and are scaled between 0 and 1 as per previous studies[29,31,84].

## Variation in social bond strength

We calculated each focal male's coefficient of variation (CV) based on its SRI values to other males. This is because normalised node strength does not contain information on a male's individual social bonds, such as whether a male mostly forms bonds of similar strengths (low CV) or stronger bonds to a small number of others (high CV).

## Mean male group size

We calculated the mean male group size for males based on the mean number of other males present during sightings of our focal males. Each male's mean group size was calculated until the year they were sampled. We applied the same restrictions as for the calculation for the SRIs (present in first 5 min, no resights, no foraging surveys). Males that were observed fewer than 15 times contributed to the mean male group size estimate. Male sex was confirmed genetically for animals that have been sampled as described in ref. 38, via observation of male-specific behaviours or visual observation of the genital region.

## Statistical analyses

We employed a two-step model selection process using linear mixed-effects models. First, we determined the optimal random effects structure by comparing four models: a null model and models including dolphin ID, year sampled, or both as random variables. We used Akaike Information Criterion (AIC) to compare models, selecting the model with the lowest AIC as the best random variable structure. Next, we conducted fixed effects selection by constructing a series of models with increasing complexity, starting from the null model and progressively adding cumulative social bond strength, male group size, chronological age, the CV of cumulative social bond strength, and the interaction between cumulative social bond strength and male group size. Like in the first step, we selected the model with the lowest AIC as the best-performing model. All models were fitted using the lme4 package v1.1-35.5[85].

To increase confidence in our findings, we also ran the best-performing model on adult males only (≥14 years old) and, separately, animals where chronological age was known with an accuracy of ±6 months. Moreover, we also generated epigenetic age estimates using seven previously described epigenetic clocks as response variables. Lastly, we also ran the best-performing model, including AgeAccel instead of epigenetic age as the response variable. AgeAccel is a commonly used measure of age acceleration in epigenetic clock studies[86]. It is defined as the residuals of a linear regression of epigenetic age on chronological age[33] and is also known as 'extrinsic age acceleration'[87]. In essence, AgeAccel represents how much older or younger an individual is epigenetically compared to what would be expected based on their chronological age[15].

Partial residual plots (Fig. 2) were used to visualise the relationships between predictors and epigenetic age while accounting for the effects of all other variables in the model[88,89]. Partial residuals were calculated by subtracting the fitted effects of all other predictors from the observed response values, leaving only the component of variation explained by the predictor of interest plus the residual variation. This approach ensures that both the plotted data points and regression lines represent the same statistical

relationship (the partial effect of each variable while controlling for others), thereby avoiding potential visual confusion that can arise when raw data points show different patterns than model-predicted effects. For comparison, an alternative visualisation showing raw observed data values plotted with model-predicted relationships is presented in Supplementary Fig. 4.

All analyses were conducted in R v.4.4.0[90].

## Reporting summary

Further information on research design is available in the Nature Portfolio Reporting Summary linked to this article.

## Data availability

The epigenetic data used for this project is part of the data release from the Mammalian Methylation Consortium (https://clockfoundation.org/MammalianMethylationConsortium) and is also posted on the Gene Expression Omnibus website (complete dataset: GSE223748).

## Code availability

Code for epigenetic clock calibration and the statistical analyses described in this study is publicly available on https://github.com/ligerber/epiclock. The mammalian methylation array is available from the non-profit Epigenetic Clock Development Foundation.

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

## Acknowledgements
Data collection for this research was carried out on Gathaagudu, Malgana Sea Country, and we acknowledge the traditional custodians of the region. We thank William B. Sherwin, all volunteers, RAC Monkey Mia Dolphin Resort, Monkey Mia Wildsights, and the local Department of Biodiversity, Conservation and Attractions (DBCA) staff for their continued support and assistance. L.G. was supported by an Early Postdoc Mobility Grant from the Swiss National Science Foundation (P2ZHP3_200011) and received funding for sequencing from the Claraz Foundation. S.L.K. was supported by The Branco Weiss Fellowship—Society in Science. L.A.R. was supported by the University of New South Wales Scientia programme. K.G.H. was supported by an Australian Government Research Training Program Scholarship at the University of Western Australia. K.J.P. was supported by a Postdoc Grant from the University of Zurich. Further funding was received from the Swiss National Science Foundation (31003A_149956 to MK), the A.H. Schultz Foundation (MK, LG), the Australian Research Council A19701144, DP0346313), the Eppley Foundation for Research, the Seaworld Research and Rescue Foundation, and the W.V. Scott Foundation.

## Author contributions
Conceptualisation: L.G., M.K., and L.A.R. Methodology: L.G., K.J.P., E.P.W., and O.F., Investigation: L.G., K.J.P., M.K., R.C.C., S.J.A., S.L.K., and K.G.H., Formal analysis: L.G., E.P.W., A.M.K., and O.F., Resources: M.K., L.A.R., R.C.C., S.J.A., S.L.K., and L.G., Data curation: S.L.K., M.K., R.C.C., and S.J.A., Writing—original draft: L.G. and L.A.R. Writing—review & editing: L.G., K.J.P., S.L.K., K.G.H., S.J.A., R.C.C., A.M.K., E.P.W., O.F., M.K., and L.A.R.

## Competing interests
The authors declare no competing interests.
