## [Transparent Peer Review file · Communications Biology]

Social bonds decrease epigenetic age in male bottlenose dolphins

Corresponding Author: Dr Livia Gerber

This manuscript has been previously submitted at another journal. This document only contains information relating to versions considered at Communications Biology.

Version 0:

Reviewer comments:

Reviewer #1

(Remarks to the Author)

Gerber et al. present research showing a negative relationship between strong social bonds and biological age (demonstrated using an epigenetic clock). Further, this relationship is not related to social group size, as that has a positive relationship with biological age. This same pattern has been identified in humans, however using wild social animals indicates the pattern of sociality on biological age may be more universal across social mammals.

My biggest question, and one that makes it a little difficult to assess the results of the paper, is how the analyses were done. I read through both the methods and the results and need more information on how the models were constructed. To the point, looking at Figure 2A (thank you for including the raw data, by the way!), I find the negative relationship shown unconvincing given the data presented (not impossible with repeat sampling, but I need a lot more information about how the models were generated as well as the samples themselves to determine if such a pattern is believable). This figure determines the main findings of the paper and I would like to better understand it.

In the introduction, I was surprised there wasn't more information on the relationship between sociality and biological age in humans.

Also, for the 10 individuals who were samples at multiple ages- I'm confused how the behavioral data were computed; was each sample treated individually, or did all samples have the same behavioral values?

Finally, I'm also curious, if the argument is for using biological clocks, what is the relationship between chronological clock and social bonds? Is biological clock a better indicator than actual age? I feel like understanding this might have some implications for the interpretation of the data as a whole.

Minor comments

Introduction, 4th paragraph- I'm not sure "interrogation" is the correct word here.

Introduction, 5th paragraph- what does "homogenous social bonds" mean?

Figure 1- not a huge issue, but I found it a little confusing that the axes aren't equivalent (I think it would be easier to interpret if they were).

Discussion, 2nd paragraph- on the flip side, greater social interactions also leads to greater disease spread. How might this influence the results?

Discussion, last paragraph- although the authors argue that their findings do not permit conclusions on the causality between ageing and social bonds (I agree), some postulating on how that might arise would not be amiss.

Please double check the methods- since they're spread out between the results, methods, and supplemental sections, I found a few discrepancies (e.g. number of individuals used) and it's a bit confusing to follow.

Reviewer #2

(Remarks to the Author)

I enjoyed reading this paper very much: it is solid approach to addressing this very interesting questions. It also has broad-ranging implications: the results are very relevant to others studying cetaceans, but also the wider community of researchers studying (or interested in) the impacts of stress and other characteristics of life on the ageing process. Having said that, I do have some minor concerns. Specifically, I think that some of the methods and/or data sets could use a bit more explanation and justification. Second, I would also like to see some of the analyses revised to include a more complete assessment of the effects of some of the predictor variables on epigenetic age. Such analyses are, I feel, necessary for proper interpretation of the results.

SPECIFIC COMMENTS

1. Results: Figure 1 is very helpful, but it would also be very informative for readers to know the results as they relate to the individuals sampled at multiple ages. Adding this information to Figure 1 may make it too busy, but perhaps having a table with these individuals, their chronological ages, and their biological ages for these different time points would work well. I think showing this aspect of the validation of this approach is very important.
2. Lines 122-123: I assume that these "focal" individuals were seen at least 15 times once they were already adults and in established social groups. Is that correct? Either way, more information on these individuals is required here for the readers to really understand the context of these individuals. It would be very different if these individuals were seen 15 times throughout their life (including as calves) versus 15 times since they were over the age of, say, 10. The Methods say that this count can include their first sighting (i.e., as calves). However, the Introduction (and other literature) suggests that their social bonds "crystallise" after a decade. Thus, it seems like only using social data for males ≥ 10 years would be appropriate. What are the patterns observed if only such males are included? I would suggest three things: (1) be more explicit about the ages (at time of sampling) for the individuals used in this part of the analyses, and perhaps include a table (in the SI) of the ages of these individuals for their observations contributing to their bond strength score; (2) if some were ≤ 10 during some of the observations used to quantify their social bond strength, justify their use in such an analysis; and (3) also include analyses using only individuals whose social bond strength only includes observations when they were ≥ 10 years old.
3. Lines 160-163: It would be helpful if the authors provided and discussed the effect size here, and the biological meaning of it. For example, am I correct in interpreting the coefficient as - a male with a mean social bond strength of 1 is expected to have a biological age ~ 5 years younger than a same-aged male with a social bond strength of 0? Converting the statistical results to biological implications is important for work like this so that readers can differentiate between statistical significance and biological significance.
4. Lines 173-174: This sentence doesn't seem formatted correctly.
5. Results: It would be helpful to see a plot of epigenetic versus social bond strength, and versus group size, independently; as well as a plot of social bond strength versus group size. Looking at all four of these plots together (the three pairwise comparisons, and the combined model) could greatly improve the readers' understanding and interpretation of the data. My concern is that when all of these are thrown into one model, the authors are committing a "Table 2 fallacy" when they interpret the coefficients (e.g., Westreich & Greenland 2013). In the terms of causal inference, I am specifically concerned that social bond strength is a mediator between group size and epigenetic age. If so, then the signal from each could be interfering with each other in the combined model (e.g., McElreath 2020, Rohrer 2018).

McElreath R (2020) *Statistical Rethinking*, 2nd Edition. CRC Press, Boca Raton, FL.

Rohrer JM (2018) Thinking clearly about correlations and causation: Graphical causal models for observational data. *Advances in Methods and Practices in Psychological Science* 1(1): 27-42.

Westreich D, Greenland S (2013) The Table 2 Fallacy: Presenting and interpreting confounder and modifier coefficients. *American Journal of Epidemiology* 177(4): 292-298.

Reviewer #3

(Remarks to the Author)

This study offers an interesting investigation into the relationship between sociality and epigenetic age in a non-model system (dolphins). However, to ensure the manuscript meets the journal's standards for publication, I recommend the following revisions:

Introduction

The introduction is well written and although the goal of the manuscript is clear, the introduction itself lacks a clear rationale beyond a simple comparison with other social groups. The discussion (lines 173-178) offers compelling arguments for the potential observation of decreased biological age in social species. To strengthen the manuscript, this discussion section should be incorporated into the introduction to establish a clear hypothesis and theoretical framework. This will provide a more robust foundation for the research question. Also it is important to be considered in the introduction section, that because

age determination through genome methylation (epigenetics) is a relatively new topic, it would be beneficial to include a brief paragraph that provides a deeper explanation, particularly regarding non-model species systems. While I understand that this may not be the primary focus of the manuscript and that this information is available in other works, a concise overview would greatly aid readers who may not have a background in genetics.

Material and Methods

The materials and methods section is well articulated, addressing all critical aspects needed to conduct the study. However, I would like to highlight a few points that I believe warrant further explanation from the authors:

1. Minimum Chronological Age: What was the minimum chronological age considered for the analysis? Understanding that the strength and number of social relationships may change over time, as noted in the introduction. It is also important to define the minimum age of sexual maturity for an individual to be recognized as a potential father, as it is expected a potential link between social bonds and reproductive success.
2. Social connectivity: The manuscript states that only individuals seen at least 15 times during the study period, with encounters more than two hours apart, were included in the analysis. However, it is unclear how the authors normalized this data to prevent bias, for example, treating an individual seen 20 times in one year the same as an individual seen only once a year but for 20 consecutive years.
3. Estimation of Average Male Group Size: How did the authors estimate the average group size of males? What methods were used to determine that all individuals in a given group were indeed male, or how did they ascertain the number of males present if not all were sampled?
4. Genetic Methods: Please mention or reference Peters, Gerber et al. (2023) regarding the sampling methods and the genetic protocols for sample storage, DNA extraction, and related procedures.
5. Mention how specimens were individually identified (how many microsats and PID/PID-sibs estimates)?
6. I didn't find the literature from 44 onwards.

Results

The results are brief and concise, but lack crucial details for clarity. While the figure shows a trend and the authors report statistically significant results for social bonds and group size, the level of significance (p-values) and regression coefficients are missing. The trend lines also exhibit subtle slopes, making the regression coefficients essential for proper interpretation. Furthermore, the analysis omits a crucial section on the interaction between group size and social bonds within a single model. This interaction term would clarify which factor (strong peer relationships or larger group size) has a greater influence on epigenetic age estimation. Finally, did the group size remain consistent for each individual throughout the sampling period, or did it vary? This individual-level variation in group size could explain the observed positive relationship between group size and biological age.

Discussion

The discussion is interesting but requires significant expansion. Before interpreting the results, the authors should address fundamental ecological and demographic factors of the Shark Bay dolphin population, including: population size, social dynamics and behavior (e.g., fission-fusion patterns), capture-recapture rates (to assess the population residency patterns), presence of predators, and estimated lifespan. Addressing these points will allow for a more nuanced interpretation of the findings. Also, while comparisons to human studies are insightful, the evolutionary forces do not act equally in humans and wild species populations, thus the discussion should warrant a more cautious approach. This comparison should be removed or significantly revised. Finally, the discussion needs to explicitly address the relative importance of small social group membership versus the maintenance of established social bonds in influencing epigenetic age. This could be fruitfully discussed from both social evolutionary and ecological perspectives, considering resource competition (females, food) and the shelter effect of pertaining to larger groups.

Version 1:

Reviewer comments:

Reviewer #2

(Remarks to the Author)

I appreciate all of the work that the authors have put into making the revisions. They have done a good job at addressing my comments and those of the other reviewers. However, the manner in which they clarified their analyses and visualized the data more has unfortunately created one more issue.

Specifically, in Figure 2 they plot the original data points (in black) and the model-estimated individual effects of each of three variables in the blue regression lines and intervals. This is very disorienting and unnecessarily confusing. Specifically, panel B is the most dissonant: a regression through the points would clearly have a positive slope, whereas they are

showing a negative regression line. I understand that BOTH of these are true (that there would be a positive relationship *without accounting for the other variables* - hence the pattern with the points, but that there is a negative effect of bond strength *once you account for the other variables*). However, given that the whole paper is focused around the interpretation of the negative relationship between bond strength and epigenetic age, this figure should be clearer, otherwise I can see it generating problems for most readers. I therefore very strongly recommend that the authors change these panels to make their message more clear. There are two ways in which they can do this. Perhaps the easiest way would be to remove the "observed" points from each panel, and just show the trend lines and uncertainties for the individual effects of each variable (while accounting for the others). The second option would be to change the points in each panel so that BOTH the points and the regression lines show the predicted relationships between each variable and epigenetic age while accounting for the effects of the other variables. Either of these options would make these panels more clear, and would strengthen the authors' case for their interpretation of the data.

Reviewer #3

(Remarks to the Author)

Dear authors,

Thank you for your recent revisions. I am pleased to see that the manuscript has improved significantly following the incorporation of the comments and suggestions of the three reviewers. At this stage, I have no further questions or comments.

Congratulations on your work, and I appreciate your efforts in addressing the previous feedback.

Version 2:

Reviewer comments:

Reviewer #2

(Remarks to the Author)

The authors have done a very good job addressing my remaining concerns (those associated with the presentation of results in Figure 2). I think that the manuscript is much stronger now, and I have no further comments. It is exciting and interesting work, and I look forward to seeing it published.

Reviewers' comments:

Reviewer #1 (Remarks to the Author):

Gerber et al. present research showing a negative relationship between strong social bonds and biological age (demonstrated using an epigenetic clock). Further, this relationship is not related to social group size, as that has a positive relationship with biological age. This same pattern has been identified in humans, however using wild social animals indicates the pattern of sociality on biological age may be more universal across social mammals.

My biggest question, and one that makes it a little difficult to assess the results of the paper, is how the analyses were done. I read through both the methods and the results and need more information on how the models were constructed. To the point, looking at Figure 2A (thank you for including the raw data, by the way!), I find the negative relationship shown unconvincing given the data presented (not impossible with repeat sampling, but I need a lot more information about how the models were generated as well as the samples themselves to determine if such a pattern is believable). This figure determines the main findings of the paper and I would like to better understand it.

We thank Reviewer 1 for their detailed and insightful comments on our manuscript. We acknowledge that our initial manuscript did not contain enough information on model construction to replicate our findings in a convincing manner. Our revised manuscript contains detailed information on model construction, as well as further analyses including a second measure on age acceleration (AgeAccel) to support our findings. The manuscript now describes the implementation of a comprehensive model comparison and selection procedure, where we compare multiple different model constructions using both Akaike's Information Criterion (AIC). During this model comparison process, we first determined the optimal random effects structure. Second, we compared different configurations of direct and interaction effects and selected the best performing model on the basis of AIC. This approach is now described on lines 395 – 404. Detailed information on the model comparison and AIC values is provided in the SI in Tables S1-S2. Our revised manuscript also contains additional information on the samples themselves. This can be found in Data S1.

In the introduction, I was surprised there wasn't more information on the relationship between sociality and biological age in humans.

It indeed appears surprising that there is not much information on the relationship between sociality and biological age in humans, when a plethora of studies have investigated the influence of sociality on lifespan in humans and social mammals. However, a reliable biomarker for biological age was unavailable until epigenetic clocks were developed a decade ago. Most studies applying epigenetic clocks investigated the influence of various diseases and behaviours (e.g., smoking) that were known to be associated with lifespan before turning to social relationships in the last few years. To the best of our knowledge, we cite these four studies in the manuscript on line 84.

Also, for the 10 individuals who were samples at multiple ages- I'm confused how the behavioral data were computed; was each sample treated individually, or did all samples have the same behavioral values?

Each sample was treated individually. This is now made clearer (see lines 359 - 360):

“For repeatedly sampled individuals, this meant that social connectivity was calculated multiple times, once per sampling point.”

Finally, I'm also curious, if the argument is for using biological clocks, what is the relationship between chronological clock and social bonds? Is biological clock a better indicator than actual age? I feel like understanding this might have some implications for the interpretation of the data as a whole.

To address this question, it is crucial to distinguish between chronological age, i.e., the time elapsed since birth, and biological age, often measured through epigenetic clocks, which reflects an individual's physiological age and can differ from its chronological counterpart. This difference is now defined clearer in the revised manuscript on lines 67 - 71:

“However, for a comprehensive understanding of the interplay between social factors and age, we also need to consider biological age. Biological age reflects an individual's physiological age and serves as an indicator of overall health and ageing status. Consequently, biological age can differ from chronological age within an individual and between individuals of the same chronological age.”

Social bonds change with chronological age due to changing needs and experiences at different life stages. For example, juveniles might form bonds that are essential for learning skills required in adulthood, while adults may forge social bonds that are crucial for reproductive success or survival.

Biological age captures the cumulative effects of various factors such as environmental conditions, stress exposure, and overall health on the aging process. These factors directly influence an organism's physiological state and, consequently, its capacity to form and maintain social bonds. Thus, biological age can explain discrepancies in social bonding capabilities among individuals of the same chronological age, offering a more comprehensive understanding of the interplay between social relationships and the rate of ageing.

Given that social bonds are influenced by both chronological and biological age, we correct for chronological age to ensure its influence is accounted for in our study. This allows us to isolate the effect of social variables on biological age independent of chronological age. We have added this to the revised manuscript on lines 112 - 116 to improve clarity:

“..we employed a species-specific epigenetic clock while controlling for chronological age to isolate the effect of social variables on biological age as suggested by Krieger et al.. Further, controlling for chronological age is crucial because social factors can change over an animal's lifespan, and thus with chronological age.”

In essence, biological age provides insight into an individual's physiological capacity for social engagement, while chronological age remains relevant in determining life stages and broader patterns of social behaviour in a species. Therefore, both chronological and biological age play important but complementary roles in understanding sociality.

Minor comments

Introduction, 4th paragraph- I'm not sure "interrogation" is the correct word here.

Changed to 'inference'.

Introduction, 5th paragraph- what does "homogenous social bonds" mean?

Modified to 'males that invest more equally in social bonds with other males'.

Figure 1- not a huge issue, but I found it a little confusing that the the axes aren't equivalent (I think it would be easier to interpret if they were).

We have made the axes of Figure 1 depicting the correlation between epigenetic and chronological age equivalent for easier interpretability.

Discussion, 2nd paragraph- on the flip side, greater social interactions also leads to greater disease spread. How might this influence the results?

We thank Reviewer 1 for their insightful comment. While offering increased protection from predators, group-living is linked to greater disease spread. Various diseases are known to accelerate ageing in humans. Similarly, larger group sizes in dolphins may accelerate ageing not only via resource competition over females, but also diseases. We have now included this in the discussion (lines 272 - 275):

".. group-living can increase disease risk. In humans, various diseases have been linked to accelerated ageing such as Covid-19, HIV-1, and cancer. Additional studies are required to disentangle the sheltering effect of living in larger groups that may decelerate ageing from disease risk that is linked to accelerated ageing."

Discussion, last paragraph- although the authors argue that their findings do not permit conclusions on the causality between ageing and social bonds (I agree), some postulating on how that might arise would not be amiss.

We have expanded the paragraph on the causality between ageing and social bonds. It now reads (lines 287 - 301):

"While our findings do not permit definitive conclusions on the causality between ageing and social bonds, we can hypothesise about several potential mechanisms for this relationship. Social bonds could decelerate the aging process through various pathways. For instance, strong social bonds might reduce stress as found in various social animals. Social bonds can reduce stress by improving access to resources via cooperation, or reducing energy expenditure in foraging or defence activities (Snyder-Mackler, et al. 2020). This stress reduction could lead to decreased inflammation and oxidative stress, both of which are associated with accelerated ageing. Alternatively, individuals that age more slowly might be better equipped to invest and maintain social bonds. Slower ageing is likely to be associated with better physical condition, allowing individuals to participate in social interactions including the various cooperative behaviours displayed by male dolphin alliances. These mechanisms are not mutually exclusive and could operate in a feedback loop, where strong social bonds and slower ageing reinforce each other over time, improving an individual's overall health. Future research could shed light on the mechanisms underlying the relationship between ageing and sociality, and this might be best achieved in an experimental system."

Please double check the methods- since they're spread out between the results, methods,

and supplemental sections, I found a few discrepancies (e.g. number of individuals used) and it's a bit confusing to follow.

We have double checked the numbers and found no further discrepancies.

Reviewer #2 (Remarks to the Author):

I enjoyed reading this paper very much: it is solid approach to addressing this very interesting questions. It also has broad-ranging implications: the results are very relevant to others studying cetaceans, but also the wider community of researchers studying (or interested in) the impacts of stress and other characteristics of life on the ageing process. Having said that, I do have some minor concerns. Specifically, I think that some of the methods and/or data sets could use a bit more explanation and justification. Second, I would also like to see some of the analyses revised to include a more complete assessment of the effects of some of the predictor variables on epigenetic age. Such analyses are, I feel, necessary for proper interpretation of the results.

We thank Reviewer 2 for commending us on our work and deeming our work as a solid approach to interesting questions. As suggested by Reviewer 2, we have added more details on our methods and data and included additional analyses as suggested by Reviewer 2. Overall, we deemed the suggestions of Reviewer 2 as highly valuable to strengthen our manuscript further.

SPECIFIC COMMENTS

1. Results: Figure 1 is very helpful, but it would also be very informative for readers to know the results as they relate to the individuals sampled at multiple ages. Adding this information to Figure 1 may make it too busy, but perhaps having a table with these individuals, their chronological ages, and their biological ages for these different time points would work well. I think showing this aspect of the validation of this approach is very important.

The biological and chronological ages are provided in Data S1 alongside the social variables. We mistakenly only provided the sample IDs and not the individual IDs in our first submission of the Data. This mistake has been rectified, and the individual IDs are now included. To easily compare samples of the same individuals taken at different points in time, Data S1 is ordered by individual ID.

2. Lines 122-123: I assume that these "focal" individuals were seen at least 15 times once they were already adults and in established social groups. Is that correct? Either way, more information on these individuals is required here for the readers to really understand the context of these individuals. It would be very different if these individuals were seen 15 times throughout their life (including as calves) versus 15 times since they were over the age of, say, 10. The Methods say that this count can include their first sighting (i.e., as calves). However, the Introduction (and other literature) suggests that their social bonds "crystallise" after a decade. Thus, it seems like only using social data for males ≥ 10 years would be appropriate. What are the patterns observed if only such males are included? I would suggest three things: (1) be more explicit about the ages (at time of sampling) for the individuals used in this part of the analyses, and perhaps include a table (in the SI) of the ages of these individuals for their observations contributing to their bond strength score; (2) if some were ≤ 10 during some of the observations used to quantify their social bond strength, justify their use in such an analysis; and (3) also include analyses using only individuals whose social bond strength only includes observations when they were ≥ 10 years old.

(1) We thank Reviewer 2 for their input. To clarify, the focal males in this study are not restricted to adults, and the males span an age range between 0 and 29 years old. This is now included in the manuscript (lines 153 - 154). The revised version also clearly states that the inclusion of a wide age range is accounted for by controlling for

chronological age in our analyses (lines 112 – 116). Please also see our reply to Reviewer 1 on the difference between chronological and biological age. Moreover, the Reviewer's comment has made us realise that we did not refer to the dataset containing information on the focal animals and samples (Data S1) sufficiently and further, that some information in there was missing. Our revised manuscript now refers to Data S1 in three locations (lines 141, 157, 370). Data S1 now also includes each animal's year of birth and the year of an animal's first non-foraging survey and its last non-foraging survey before sampling (typically the year the sample was collected). This demonstrates that for most males, we have seen them throughout their life, and this permits the readers to infer the timespan over which the social variables were calculated. This is now also included in the manuscript (lines 368 - 370) and reads:

"For all animals with a minimum of 15 sightings, the timespan and thus age range over which the social variables are calculated are provided in Data S1."

We acknowledge that our social network metrics are based on observations over a wide time span, which can include earlier life stages. However, we deem this approach justified for various reasons: Research on both sexes in this population has shown that social traits are stable across the lifetime (Evans, et al. 2021). This stability suggests that early-life social patterns are indicative of long-term social tendencies, as also observed in other species such as chimpanzees and starlings, for example (Koski 2011; Thys, et al. 2017). Moreover, we indirectly correct for the timespan over which an animal was seen by controlling for chronological age, two highly correlated variables.

- (2) It is the formation of male alliances that takes almost a decade to crystallise, but not necessarily the social bonds per se. Moreover, this decade does not refer to an animal's first decade of life, but when animals are between 8 years and 14 years old. In contrast to earlier studies investigating the maintenance of individual social relationships with potential future alliance partners (Gerber, et al. 2019), our current study focuses on an individual's overall social bond strength. The stability of social traits across lifespans (Evans, et al. 2021) and the predictive power of juvenile social patterns for adult social networks (Holmes, et al. 2024) support our approach of including earlier life stage data. This allows for a more comprehensive picture on the interactions between social bonds and ageing while maximising sample size.*
- (3) To address the reviewer's suggestion, we have run an additional analysis including data of males sampled as adults (≥ 14 years of age) only. Social bond strength remained a significant predictor of epigenetic age but not group size. We have added details on the analysis and results to lines 191, 202 – 203, and 405 – 406) in the main manuscript and to Table S4 in the Supplementary Information. Unfortunately, we lack the sample size to carry out this analysis with social network metrics calculated with sightings during adulthood only.*

3. Lines 160-163: It would be helpful if the authors provided and discussed the effect size here, and the biological meaning of it. For example, am I correct in interpreting the coefficient as - a male with a mean social bond strength of 1 is expected to have a biological age ~5 years younger than a same-aged male with a social bond strength of 0? Converting

the statistical results to biological implications is important for work like this so that readers can differentiate between statistical significance and biological significance.

Thank you for your helpful feedback. We have revised this paragraph as follows to be more specific about the size of these effects, and have used more precise statistical description to exercise caution around causal interpretation of association effects, as discussed regarding your other comments about causality below. We have also provided standardised regression coefficients alongside the raw coefficients in Table 1, expressing the predicted effects in terms of a 1 standard deviation change in the predictor, and included 95% confidence intervals, in order to provide more detail and clarity about the magnitude and biological significance of these relationships (lines 162 – 169):

“The selected LMM revealed a statistically significant negative effect of cumulative social bond strength, i.e., social connectivity, on epigenetic age (Table 1, Fig. 2), and a significant positive relationship with mean male group size in male dolphins. A 1 standard deviation (0.17) increase in cumulative social bond strength is associated with a 1.69 years lower epigenetic age (95% CI: -3.03, -0.35), while a 1 standard deviation (1.44) increase in male group size is associated with 1.33 years higher epigenetic age (0.27, 2.40), while accounting for the other fixed and random effects included in our main model.”

4. Lines 173-174: This sentence doesn't seem formatted correctly.

We have corrected the formatting of this sentence (lines 248 - 249):

“Social factors and health are highly interlinked in social mammals: social adversity increases the risk of multiple diseases and death (reviewed in ⁵).”

5. Results: It would be helpful to see a plot of epigenetic versus social bond strength, and versus group size, independently; as well as a plot of social bond strength versus group size. Looking at all four of these plots together (the three pairwise comparisons, and the combined model) could greatly improve the readers' understanding and interpretation of the data. My concern is that when all of these are thrown into one model, the authors are committing a "Table 2 fallacy" when they interpret the coefficients (e.g., Westreich & Greenland 2013). In the terms of causal inference, I am specifically concerned that social bond strength is a mediator between group size and epigenetic age. If so, then the signal from each could be interfering with each other in the combined model (e.g., McElreath 2020, Rohrer 2018).

McElreath R (2020) Statistical Rethinking, 2nd Edition. CRC Press, Boca Raton, FL.

Rohrer JM (2018) Thinking clearly about correlations and causation: Graphical causal models for observational data. *Advances in Methods and Practices in Psychological Science* 1(1): 27-42.

Westreich D, Greenland S (2013) The Table 2 Fallacy: Presenting and interpreting confounder and modifier coefficients. *American Journal of Epidemiology* 177(4): 292-298.

We appreciate your thoughtful comments and suggestions for improving our analysis and presentation of results. We have addressed your concerns and implemented the suggested changes. Here's our response to your specific points:

1. *Additional plots: We have created the plots you suggested, showing the pairwise relationships between epigenetic age, social bond strength, and group size, as well as a combined model plot. These plots are now included in our revised*

Supplementary Information (Fig S2). We agree that these visualisations enhance the readers' understanding of the relationships between our key variables.

2. *Addressing the "Table 2 fallacy" and potential mediation effect: We acknowledge the concern about the potential for misinterpretation when presenting coefficients from a combined model (Westreich & Greenland, 2013). To address this, we have been careful to use precise language for interpreting regression coefficients, emphasising that these represent conditional associations rather than total effects. We also appreciate your insight regarding the possibility that social bond strength might mediate the relationship between group size and epigenetic age. To address this concern, we have conducted a mediation analysis and presented brief methods, results (Table S4) and interpretation in the Supplementary Information on pages 7 – 8.*
3. *We have also expanded the Discussion to address the possibility of more complex causal relationships which are beyond the scope of the present analysis, and suggest alternative modelling frameworks which could be explored in future studies where moderation or mediation relationships are thought to be present (lines 171 – 181):*

"Given the possibility of more complex causal pathways than those captured in our principal analyses, and the 'table 2 fallacy' risk of potentially over-interpreting regression coefficients from mixed effects models, we conducted a mediation analysis to assess whether social bond strength mediates the relationship between male group size and epigenetic age acceleration. Our mediation analysis (see Supplementary Information and Table S4) found no significant mediation effects. This suggests that while both group size and social bonds independently influence epigenetic ageing, the causal pathway is not well captured by a simple mediation relationship. Future studies with larger sample sizes could employ more sophisticated analytical frameworks such as structural equation modelling to disentangle the potentially complex causal interrelationships between social structure, bond formation, and biological ageing processes."

We believe these changes provide a more comprehensive and nuanced presentation of our findings, addressing the concerns about the "Table 2 fallacy" and potential mediation effects. We have strived to present our results in a way that allows readers to understand the complex relationships in our data while acknowledging the limitations of our analytical approach.

Reviewer #3 (Remarks to the Author):

This study offers an interesting investigation into the relationship between sociality and epigenetic age in a non-model system (dolphins). However, to ensure the manuscript meets the journal's standards for publication, I recommend the following revisions:

Introduction

The introduction is well written and although the goal of the manuscript is clear, the introduction itself lacks a clear rationale beyond a simple comparison with other social groups. The discussion (lines 173-178) offers compelling arguments for the potential observation of decreased biological age in social species. To strengthen the manuscript, this discussion section should be incorporated into the introduction to establish a clear hypothesis and theoretical framework. This will provide a more robust foundation for the research question. Also it is important to be considered in the introduction section, that because age determination through genome methylation (epigenetics) is a relatively new topic, it would be beneficial to include a brief paragraph that provides a deeper explanation, particularly regarding non-model species systems. While I understand that this may not be the primary focus of the manuscript and that this information is available in other works, a concise overview would greatly aid readers who may not have a background in genetics.

We thank Reviewer #3 for considering our introduction to be well written with a clearly recognisable goal. It appears however, that the rationale for our study was not clearly discernible in our initial manuscript. In our revised manuscript, we now clearly state that not much is known on the relationship between epigenetic age and social factors and that filling this gap of knowledge will have far-reaching implications for both evolutionary biology and health.

As requested by the Reviewer, our revised manuscript contains additional information on epigenetic clocks in non-model organisms (lines 86 – 95):

*“Only a few studies have investigated the effect of social variables on the rate of ageing outside of humans. The results have been mixed: while high social status was linked to accelerated ageing in male adult baboons (*Papio cynocephalus*), social bond strength in female baboons did not explain variation in ageing. Similarly, pair-bonding in prairie voles (*Microtus ochrogaster*) did not appear to affect the rate of ageing. Given these mixed results across species and the limited number of studies, the relationship between social bonds and biological ageing requires further investigation. This is crucial not only for comparative purposes, but also to shed light on how social factors may buffer against stress, and thus protect against accelerated ageing, as well as to gain insights into the evolutionary trade-offs between social behaviour, resource allocation, and longevity.*

We thank the Reviewer for their suggestions to improve clarity of the introduction and thereby the aims of our study.

Material and Methods

The materials and methods section is well articulated, addressing all critical aspects needed to conduct the study. However, I would like to highlight a few points that I believe warrant further explanation from the authors:

1. Minimum Chronological Age: What was the minimum chronological age considered for the analysis? Understanding that the strength and number of social relationships may change over time, as noted in the introduction. It is also important to define the minimum age of sexual maturity for an individual to be recognized as a potential father, as it is expected a potential link between social bonds and reproductive success.

For this study, we considered animals of all ages, as long as they met the criterion of having been sighted at least 15 times. In this study, we are interested in the effect of social bonds on biological ageing independent of chronological age. We acknowledge that we did not sufficiently accentuate the difference between chronological and biological age in our initial submission but have corrected this in our revised manuscript (lines 67 – 71, please also see our response to Reviewer 1 on the difference between chronological and biological age). On lines 113 – 116 we are now also clearly stating that

“.. controlling for chronological age [allows] to isolate the effect of epigenetic age on social variables. Further, controlling for chronological age is crucial because social factors can change over lifespan and thus with chronological age.”

We would also like to refer to our comment to Reviewer 2, where we point out that social traits are usually stable over an individual’s lifetime (i.e., highly social animals usually remain highly social throughout their lives, even when they lose or accrue social bonds with chronological age). This information is now also included in our revised manuscript (lines 116 - 118).

The link between social bonds and reproductive success in males has been investigated in depth in a previous publication (Gerber, et al. 2022). Here, we are interested in the relationship between social bonds and ageing across lifespan. We agree, however, that similar to the study on male baboons referred to in the introduction (Anderson, et al. 2021), investigating the effect of male-male competition on ageing in males would make an interesting future study.

2. Social connectivity: The manuscript states that only individuals seen at least 15 times during the study period, with encounters more than two hours apart, were included in the analysis. However, it is unclear how the authors normalized this data to prevent bias, for example, treating an individual seen 20 times in one year the same as an individual seen only once a year but for 20 consecutive years.

Reviewer 2’s comment 2 is similar and we would like to refer to our reply to their comment. In sum, we acknowledge that our social network metrics can be based on observations over a wide time span. However, we deem our approach justified because social traits in this dolphin population are stable across lifetime (Evans, et al. 2021) and we control for chronological age and thus, the timespan an animal was observed. In previous studies, we have included the years an animal was observed in as random variable. We did not do this in our current study because the number of years an animal was observed in is highly correlated with chronological age (see Data S1) which is already included in our model.

3. Estimation of Average Male Group Size: How did the authors estimate the average group size of males? What methods were used to determine that all individuals in a given group were indeed male, or how did they ascertain the number of males present if not all were sampled?

During behavioural surveys, we photograph all group members encountered. Standard photo-identification tools based on dorsal fin modifications permit individual identification of group members. As we have been studying this population for > 40 years, most animals encountered during surveys in the core study area and their sexes are known (we have detailed data on 1,333 [618 males, 715 females] individuals). Male sex is assigned based on genetics (presence of X and Y chromosomes), male-specific behaviours (e.g., herding of females), or visual observation of genitals. We have included this in the revised manuscript in the section on mean male group size calculation (lines 316 - 317):

“Group composition during observational surveys was assessed using standard photo-identification methods based on inter-individual dorsal fin differences.”

4. Genetic Methods: Please mention or reference Peters, Gerber et al. (2023) regarding the sampling methods and the genetic protocols for sample storage, DNA extraction, and related procedures.

We are grateful that Reviewer 3 noticed that crucial information on sampling methods and molecular methods was missing. All information requested above is available in the revised manuscript in either the ‘Behavioural data and tissue sample collection’ or ‘Epigenetic age’ section (starting on line 325).

5. Mention how specimens were individually identified (how many microsats and PID/PID-sibs estimates)?

Individual dolphins are individually identified using photo-identification. Photo-identification is a well-established and highly accurate method to identify small cetaceans on an individual level (Würsig and Jefferson 1990). Moreover, it is a less invasive and time and cost saving alternative to genetic identification. To the best of our knowledge, all long-term projects on bottlenose dolphins use photo identification and not genetic methods to differentiate between individuals.

The usage of photo identification to distinguish individuals is now included in the manuscript (lines 316 – 317, see our reply to point 3).

6. I didn't find the literature from 44 onwards.

We are sorry that our references were incomplete in the first submission and thank the reviewer for noticing. We have corrected this in our revised manuscript.

Results

The results are brief and concise, but lack crucial details for clarity. While the figure shows a trend and the authors report statistically significant results for social bonds and group size, the level of significance (p-values) and regression coefficients are missing. The trend lines also exhibit subtle slopes, making the regression coefficients essential for proper interpretation. Furthermore, the analysis omits a crucial section on the interaction between group size and social bonds within a single model. This interaction term would clarify which factor (strong peer relationships or larger group size) has a greater influence on epigenetic age estimation. Finally, did the group size remain consistent for each individual throughout the sampling period, or did it vary? This individual-level variation in group size could explain the observed positive relationship between group size and biological age

We have removed the Figure Reviewer 3 is referring to in the above comment, which is why our response does not refer to this section of the comment.

In our revised manuscript, we have included a stepwise model selection (see our first reply to Reviewer 1). One of the models investigated also included the interaction between group size and social bonds. The model including the interaction was not the best performing model. Moreover, the interaction between group size and social bonds was not significant.

As for the social variables, we lack sufficient data to calculate group size and cumulative social bond strength at various intervals. However, because group size is inherently linked to sociality and social variables remaining constant during an individual's life, we deem our approach suitable for this study.

Discussion

The discussion is interesting but requires significant expansion. Before interpreting the results, the authors should address fundamental ecological and demographic factors of the Shark Bay dolphin population, including: population size, social dynamics and behavior (e.g., fission-fusion patterns), capture-recapture rates (to assess the population residency patterns), presence of predators, and estimated lifespan. Addressing these points will allow for a more nuanced interpretation of the findings. Also, while comparisons to human studies are insightful, the evolutionary forces do not act equally in humans and wild species populations, thus the discussion should warrant a more cautious approach. This comparison should be removed or significantly revised. Finally, the discussion needs to explicitly address the relative importance of small social group membership versus the maintenance of established social bonds in influencing epigenetic age. This could be fruitfully discussed from both social evolutionary and ecological perspectives, considering resource competition (females, food) and the shelter effect of pertaining to larger groups.

We thank Reviewer 3 for their suggestions on how to further improve the discussion to facilitate the interpretation of our results. We have now included the requested information on the Shark Bay dolphin population, mostly by expanding the paragraph on this population in the introduction (see lines 100 - 103):

“The approximately 1,500 dolphins in our study population have a lifespan of over 40 years and live in an open social network with a fission-fusion grouping pattern, overlapping home ranges, year-round residency and bisexual philopatry.”

Information on presence of predators can be found in the discussion on line 271:

*“.. tiger sharks (*Galeocerdo cuvier*) that are highly prevalent in Shark Bay..”*

We acknowledge but respectfully disagree with the reviewer's notion on evolutionary forces not acting equally in humans and wild species. While evolutionary pressures may manifest differently across species, the fundamental processes of natural selection continue to shape both human and non-human, wild animal populations.

It is important to note that humans, despite their technological and cultural advancements, remain part of the natural world and are subject to ongoing evolutionary processes (Courtiol, et al. 2012; Milot, et al. 2011). Throughout our evolutionary history, including recent times, natural selection has played a crucial role in shaping human social behaviour and life history traits. This has resulted in humans evolving to be highly social,

cooperative creatures with relatively slow life histories, traits that are shared to varying degrees with many other social species.

We consider our comparison between human studies and our findings as carefully worded without requiring removal or revision. However, we have added that more studies are needed to fully understand the relationship between ageing and sociality more broadly (see lines 244 – 247):

“This convergence across the two distant taxa underscores the potential universality of the link between sociality and ageing. However, while these findings are compelling, more studies are needed to fully understand the complex relationship between ageing and sociality.”

The revised discussion now discusses how both, intragroup male-male competition and group-living may contribute to biological ageing (lines 265 – 275):

*“Males seen in association with a higher number of males may face more male-male competition over valuable alliance partners or higher rates of intrasexual aggression by competing more often over females with non-allied males. Further, males with weaker social bonds within alliances lead more stressful lives than allies with stronger within alliance bonds. These interactions may accelerate biological ageing, as recently has been demonstrated in baboons via an epigenetic clock. Furthermore, despite offering protection against predators, e.g., tiger sharks (*Galeocerdo cuvier*) that are highly prevalent in Shark Bay, group-living can increase disease risk. In humans, various diseases have been linked to accelerated ageing such as Covid-19, HIV-1, and cancer. Additional studies are required to disentangle the sheltering effect of living in larger groups that may decelerate ageing from disease risk that is linked to accelerated ageing.”*

We thereby now provide the readers of our manuscript with a more refined discussion.

References used in response to comments

Anderson, Jordan A., et al.

2021 High social status males experience accelerated epigenetic aging in wild baboons. *eLife* 10:e66128.

Courtiol, Alexandre, et al.

2012 Natural and sexual selection in a monogamous historical human population. *Proceedings of the National Academy of Sciences* 109(21):8044-8049.

Evans, Taylor, et al.

2021 Lifetime stability of social traits in bottlenose dolphins. *Communications Biology* 4(1):759.

Gerber, Livia, et al.

2019 Affiliation history and age similarity predict alliance formation in adult male bottlenose dolphins. *Behavioral Ecology* 31(2):361-370.

Gerber, Livia, et al.

2022 Social integration influences fitness in allied male dolphins. *Current Biology* 32(7):1664-1669.e3.

Holmes, Kathryn G., et al.

2024 Juvenile social play predicts adult reproductive success in male bottlenose dolphins. *Proceedings of the National Academy of Sciences* 121(25):e2305948121.

Koski, Sonja E.

2011 Social personality traits in chimpanzees: temporal stability and structure of behaviourally assessed personality traits in three captive populations. *Behavioral Ecology and Sociobiology* 65(11):2161-2174.

Milot, Emmanuel, et al.

2011 Evidence for evolution in response to natural selection in a contemporary human population. *Proceedings of the National Academy of Sciences* 108(41):17040-17045.

Snyder-Mackler, Noah, et al.

2020 Social determinants of health and survival in humans and other animals. *Science* 368(6493):eaax9553.

Thys, Bert, et al.

2017 Exploration and sociability in a highly gregarious bird are repeatable across seasons and in the long term but are unrelated. *Animal Behaviour* 123:339-348.

Würsig, B, and TA Jefferson

1990 Methods of photo-identification for small cetaceans. In 'Individual Recognition of Cetaceans: Use of Photo-Identification and Other Techniques to Estimate Population Parameters,.(Eds PS Hammond, SA Mizroch and GP Donovan.) pp. 43-52. International Whaling Commission: Cambridge, UK.

Reviewers' comments:

Reviewer #2 (Remarks to the Author):

I appreciate all of the work that the authors have put into making the revisions. They have done a good job at addressing my comments and those of the other reviewers. However, the manner in which they clarified their analyses and visualized the data more has unfortunately created one more issue.

Specifically, in Figure 2 they plot the original data points (in black) and the model-estimated individual effects of each of three variables in the blue regression lines and intervals. This is very disorienting and unnecessarily confusing. Specifically, panel B is the most dissonant: a regression through the points would clearly have a positive slope, whereas they are showing a negative regression line. I understand that BOTH of these are true (that there would be a positive relationship *without accounting for the other variables* - hence the pattern with the points, but that there is a negative effect of bond strength *once you account for the other variables*). However, given that the whole paper is focused around the interpretation of the negative relationship between bond strength and epigenetic age, this figure should be clearer, otherwise I can see it generating problems for most readers. I therefore very strongly recommend that the authors change these panels to make their message more clear. There are two ways in which they can do this. Perhaps the easiest way would be to remove the "observed" points from each panel, and just show the trend lines and uncertainties for the individual effects of each variable (while accounting for the others). The second option would be to change the points in each panel so that BOTH the points and the regression lines show the predicted relationships between each variable and epigenetic age while accounting for the effects of the other variables. Either of these options would make these panels more clear, and would strengthen the authors' case for their interpretation of the data.

We thank the reviewer for this important observation regarding the visualisation in Figure 2. The reviewer correctly identified that displaying raw data points alongside model-predicted effects created visual confusion, particularly in Panel B where the bivariate relationship between bond strength and epigenetic age appeared positive while the model-estimated partial effect (accounting for other variables) was negative.

While the suggestion to remove the original data points is appealing, Nature requests that 'Individual data points are shown when possible, and always for $n \leq 10$ '. Thus, we have addressed this concern by implementing the reviewer's second suggested solution. We have replaced the raw data points with partial residuals that show the relationship between each predictor and epigenetic age after accounting for the effects of all other variables in the model. This approach ensures that both the

data points and regression lines represent the same statistical relationships, i.e., the partial effects of each variable, while controlling for others.

The plots in the updated Figure 2 now show:

- Panel A (Chronological Age): Shows the partial effect of age on epigenetic age, controlling for bond strength and group size
- Panel B (Social Bond Strength): Shows the partial effect of bond strength on epigenetic age, controlling for age and group size
- Panel C (Group Size): Shows the partial effect of group size on epigenetic age, controlling for age and bond strength

This visualisation approach eliminates the apparent contradiction the reviewer noted and provides a visually clearer representation of our key finding that stronger social bonds are associated with lower epigenetic age when accounting for chronological age and group size effects.

We believe this revision significantly improves the clarity of Figure 2 and strengthens the presentation of our main findings regarding the negative relationship between social bond strength and epigenetic age.

Updated Figure 2:

We have made the following addition to the Methods to clarify this approach:

“Partial residual plots (Fig. 2) were used to visualise the relationships between predictors and epigenetic age while accounting for the effects of all other variables in the model. Partial residuals were calculated by subtracting the fitted effects of all other predictors from the observed response values, leaving only the component of variation explained by the predictor of interest plus the residual variation. This approach ensures that both the plotted data points and regression lines represent the same statistical relationship (the partial effect of each variable while controlling for others), thereby avoiding potential visual confusion that can arise when raw data points show different patterns than model-predicted effects. For comparison, an alternative visualisation showing raw observed data values plotted with model-

predicted relationships is presented in Supplementary Fig. S4.”

And we have also modified the caption for Fig. 2 accordingly:

“Fig. 2. Partial effects of chronological age, social bond strength, and group size on epigenetic age in bottlenose dolphins. Each panel shows partial residuals (black dots) and model predictions (blue lines with 95% confidence intervals) representing the relationship between one predictor and epigenetic age while controlling for the effects of all other variables in the model. (A) Positive partial effect of chronological age on epigenetic age. (B) Negative partial effect of social bond strength on epigenetic age. (C) Positive partial effect of group size on epigenetic age. Partial residuals were calculated by removing the fitted effects of all other predictors from the response variable, ensuring that both the plotted points and regression lines represent the same statistical relationships. Random effects for individual dolphin ID and year sampled were included in the model but are not visualised here. See Supplementary Figure S4 for an alternative visualisation showing raw observed data values plotted with model-predicted relationships.”

Reviewer #3 (Remarks to the Author):

Dear authors,

Thank you for your recent revisions. I am pleased to see that the manuscript has improved significantly following the incorporation of the comments and suggestions of the three reviewers. At this stage, I have no further questions or comments.

Congratulations on your work, and I appreciate your efforts in addressing the previous feedback.

We thank Reviewer #3 for commending us on our revisions and are grateful for their helpful suggestions to improve our study.